# Effect of the Altitude Gradient on the Physiological Performance of Quinoa in the Central Region of Colombia

Miguel García-Parra [1,2,*] , Diego Roa-Acosta [3] and Jesús Eduardo Bravo-Gómez [3]

1   Programa de Doctorado en Ciencias Agrarias y Agroindustriales, Departamento de Ciencias Agropecuarias, Universidad del Cauca, Popayán 190002, Colombia
2   Facultad de Ciencias Agrarias y del Ambiente, Universidad Francisco de Paula Santander, Ocaña 546551, Colombia
3   Departamento de Agroindustria, Facultad de Ciencias Agrarias, Universidad del Cauca, Popayán 190002, Colombia
*   Correspondence: miguelgarciap@unicauca.edu.co

**Abstract:** The conditions of the agroecological environment play a fundamental role in the physiological performance of quinoa; however, due to the accelerated expansion of quinoa cultivation and the great diversity of cultivars present in the world, it has not been possible to study the effect that their interaction can have, which brings with it problems in productivity and even in the adaptability of cultivars. The aim of this research was to evaluate the physiological performance of seven quinoa cultivars under three altitude gradients in the central region of Colombia (cold, temperate and warm climates). The research was developed using a completely randomized design with a $3 \times 7$ factorial arrangement where the first factor corresponded to the study areas and the second factor to the selected cultivars. The results showed a highly differential performance between the phenological, physiological and compositional variables, mainly between the quinoa cultivars planted in cold climates and those established in temperate and warm climates. In this sense, the time elapsed between the phenophases, the physiological activity associated with the chlorophyll content and the quantum efficiency of photosystem II, as well as the grain yield and its protein content, are highly influenced by the cultivar and the altitudinal gradient. The results obtained support the notion that the physiological performance of quinoa depends largely on the edaphoclimatic environment by influencing different agronomic and compositional parameters of the seeds. Additionally, it was possible to identify that the evaluated quinoa cultivars were grouped into two large groups. The first group is made up mainly of the Nueva and Soracá cultivars, while the second group includes the Nariño and Puno cultivars. These four cultivars show a lower effect of the factors and their interaction on the parameters evaluated.

**Keywords:** adaptability; climate; cultivars





## 1. Introduction

The expansion of quinoa cultivation in Colombia has been increasing in recent years. However, its production has been limited to cold climate zones with grain yields that do not exceed 2 t/ha$^{-1}$ [1]. This situation has been studied in different parts of the world and, therefore, research strategies have been generated that are intended to expand quinoa production to different agroecological zones that include temperate and warm climates. Reguera et al. [2] consolidated the first study, in which they observed phenological, productive and compositional traits of three quinoa cultivars planted under different edaphoclimatic conditions, finding a strong relationship between the cultivar and the agroecological context with respect to the variables evaluated. Such a relationship is of great importance in specialized agricultural production systems, because the effect of the environment on the compositional attributes of the products obtained is recognized, so their

interaction and response has been strongly studied in crops relevant to the food industry, such as rice, wheat and maize [3–5].

When quinoa is established in different climatic regions, changes are generated in the natural conditions and adaptability of the species, mainly in climate elements related to temperature, radiation and precipitation that are determining precursors of activities such as protein stability, the synthesis of phytohormones, the transport of substances and the biological activity of the tissues, which has made it possible to evaluate quinoa in other countries in areas where other crops have not been successfully adapted [6–8].

Thus, these researches have been consolidated mainly to evaluate the response that plants have to the genotype–environment interaction, which finally manifests itself in the physiological and biochemical expression of plants [9]. However, there are many responses that quinoa can generate to a modification, whether in the genetic or environmental module or in their interaction, rapidly favoring changes of morphological, photosynthetic, productive and compositional order [10,11].

In this sense, the effect of the environment–genotype interaction brings responses related to phenotypic plasticity, which is associated with changes related to root architecture, stem height and thickness, leaf morphology, leaf area, chlorophyll content, photosynthetic yield, productivity and proximal characteristics of the grains that are in most cases related to the stability of plant tissues, the antioxidant response of the plant, osmotic activity and the synthesis of proteins that cope with stress [12–14].

In the case of the central region of Colombia, quinoa cultivars such as Tunkahuan, Soracá and Blanca Jericó were identified, which present a variation in the duration of the phenological cycles under cold weather conditions [15,16] and may be directly associated with temperature and precipitation, influencing variables such as grain yield, harvest index, biomass and phenological time, as has already been reported in other researches [17,18]. However, there is still a lack of knowledge about the effect that the altitude gradient may have on physiological activity parameters in Colombian quinoa cultivars, so our working hypothesis was that the altitude gradient has a differential effect on the physiological performance of seven quinoa cultivars.

## 2. Materials and Methods

Experiments were established in three regions during the year 2020, with spatial specifications as shown in Table 1. The experimental design was completely randomized where each experimental unit corresponded to 9 m$^2$ (3 m × 3 m) with a 3 × 7 factorial combination. The first factor corresponded to the three altitude gradients (cold (2648 m a.s.l.), temperate (1707 m a.s.l.) and warm (698 m a.s.l.) climate), and the second factor to the seven quinoa cultivars evaluated (Blanca Real, Pasankalla, Nariño, Titicaca, Puno, Soracá and Nueva), each with three replications (i.e., for a total of 21 treatments).

**Table 1.** Location and edaphoclimatic description of the experimental regions.

| Region | Geographical Position | | Climate (Caldas Lang) | Altitude (m) | Soil Type * | T (°C) | Rainfall (mm) | Planting Day |
|--------|-------------------|------------|-----------------------|--------------|-------------|--------|---------------|--------------|
| | Latitude | Longitude | | | | | | |
| Oicatá | 5°36′ N | 73°10′ W | Cold | 2648 | Association Vertic Haplustalf—Andic Dystrustepts | 12.1 | 741 | 8 April 2020 |
| Moniquirá | 5°52′ N | 73°33′ W | Temperate | 1707 | Complex Chromic Hapluderts—Typic Dystrudepts | 18.1 | 1305.9 | 14 April 2020 |
| Pauna | 5°39′ N | 74°04′ W | Warm | 698 | Typic Eustrudepts—Typic Udorthents—Humic Dystrudepts | 23.1 | 1984 | 18 April 2020 |

* Source: IGAC, (2005).

Each experimental unit corresponded to 6 rows of 3 m in length, with a planting density of 8 kg/ha$^{-1}$ and with an average planting depth of 2 cm. Sowing, weeding, harvesting and seed selection were performed manually. Quinoa cultivars were obtained from the germplasm bank of the Government of Boyacá, the AOF-JDC Agricultural Organizations and Fruits Research Group, and local markets and farmers in the central region of Colombia.

The types of soil were reported according to what was established by the IGAC [19] while the chemical characteristics were determined by soil analysis. The type of soil in Oicatá is an association of Vertic Haplustalf—Andic Dystrustepts, with a pH of 5.8, organic matter of 4.1%, and exchangeable bases (cmol/Kg) of Ca, Mg, K and Na equivalent to 2.5, 1.12, 1.07 and 0.14, respectively. The soil type in Moniquirá is a Humic Dystrudepts—Typic Dystrudepts association with a pH of 4.94, organic matter of 6.19%, and exchangeable bases (cmol/Kg) of Ca, Mg, K and Na equivalent to 4.29, 0.51, 0.34 and 0.14, respectively. The soil type in Pauna is an intro association-Typic Eustrudepts—Typic Udorthents—Humic Dystrudepts, with a pH of 6.61, organic matter of 4.26%, and exchangeable bases (cmol/Kg) of Ca, Mg, K and Na equivalent to 28, 5.1, 2.8 and 0.5, respectively.

## 2.1. Environmental Conditions

The temperature conditions during the investigation varied significantly between the three study areas. Between the cold-temperate, cold-warm and temperate-warm climates, there was a difference in temperature of 35.4%, 48.2% and 19.8%, respectively, while for precipitation there were two rainy seasons throughout the year, with a differential intensity between the zones that varied in the same order by 45.4%, 67% and 39.68% (Figure 1).

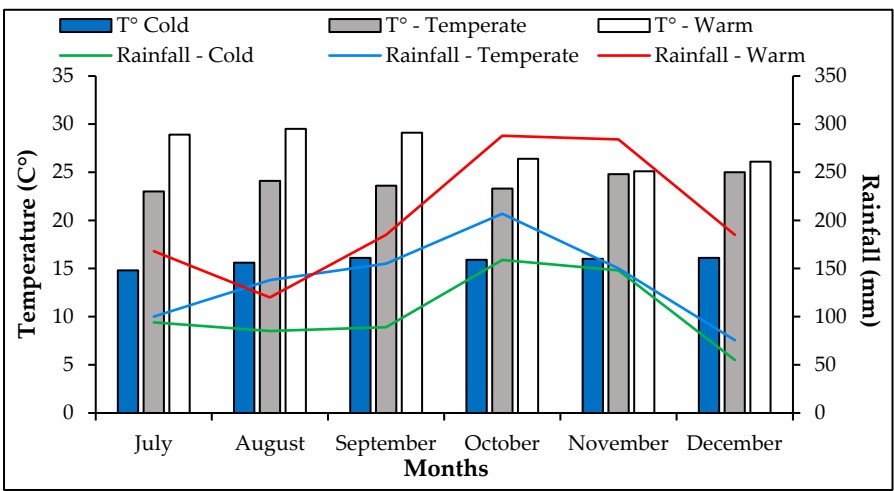

**Figure 1.** Climatic conditions of the three study regions.

## 2.2. Plant Growth and Development

The days of development were measured as a reference for the phenological phases of quinoa and based on the Biologische Bundesanstalt Bundessortenamt und Chemische Industrie (BBCH) scale, which classifies the occurrence of its phases in nine main stages of development [20].

## 2.3. Physiological Activity

The greenness index of the leaves was determined in the course of the most relevant phenological phases for quinoa together with the fluorescence of chlorophyll, reporting the maximum quantum efficiency of photosystem II (Fv/Fm), using a SPAD 502 plus chlorophyllometer (Konica-Minolta, Tokyo, Japan) and a Junior-PAM fluorometer (Walz GmbH, Effeltrich, Germany), respectively. Additionally, to determine the specific leaf area, the leaf area was measured using a LI-3000-A portable meter (LI-COR, Lincoln, NE, USA), while the stomatal density was determined by taking leaves from three thirds of the plants at the time from the grain filling phase, covering them with transparent industrial resin and later examining them in a 200X optical microscope to facilitate their quantification.

## 2.4. Grain Yield

The amount of grain per unit area was determined by manually harvesting all fully mature plants present in a linear meter. Subsequently, these were dried at room temperature

for 15 days and manually threshed to facilitate the elimination of all impurities. The seeds were placed on a digital scale to determine their weight.

*2.5. Protein Content*

The protein concentration in the quinoa cultivars under the different treatments was determined at the time of plant maturity using the Kjeldahl methodology and multiplying it by the conversion factor 6.25, which is equivalent to 0.16 g of nitrogen per gram of protein.

*2.6. Statistical Analysis*

The effects of different quinoa cultivars and their interaction with the three altitude gradients were evaluated with a two-way analysis of variance (ANOVA). The normality and data homogeneity were evaluated with the Shapiro–Wilk and Bartlett tests, respectively. Multiple analyses of means were compared using the Tukey HSD test where the probability level of $p \leq 0.05$ was considered statistically significant. The data were analyzed in the statistical program R version 3.6.1 using the Agricolae and Factorextra libraries.

**3. Results and Discussions**

*3.1. Growth and Development*

The cultivars presented different phenological times during the productive cycle, which also varied throughout all of the phenological phases. It was possible to identify that in the course of all of the phases, there were significant variations between the treatments ($p < 0.001$) due to the effect of both the cultivar and the altitudinal gradient. However, the interaction of the two factors influenced only the time to true leaves, 50% flowering and harvest ($p < 0.01$–$0.05$). Finally, the cultivar with the highest earliness was Puno in warm climatic conditions, while the latest was Soracá in cold climate (Table 2).

The quinoa cultivars showed different development speeds during the phenological cycle, allowing confirmation of the sensitivity of quinoa cultivars to the edaphoclimatic conditions of the region where quinoa is established [2]. In this sense, it is important to highlight that the late or early behavior of a quinoa cultivar is the result of the expression of the genetic character of the species and its interaction with the environment, so that the identification of cultivars with high productive potential for a specific edaphoclimatic region is an important task when establishing a productive system [21]. Some research on quinoa has shown that high rainfall favors the lengthening of the plant's life cycle, attributed mainly to the constant cellular turgor, which affects the movement of compounds that allow cell division and elongation, through which they form structures such as leaves, stems and roots [1].

For this reason, the study of the phenological development of quinoa has become relevant within its measurement through the BBCH scale, since the assignment of codes for each of the development stages favors the analysis of the growth and development of the species [20]. In this sense, the identification of the phenological phases in quinoa has been conducted under the influence of factors associated with fertilization, genetic diversity, irrigation and phenotypic characterizations [14,22,23], showing changes even under the same genetic character, which supports observations in this research.

*3.2. Total Chlorophyll Content*

As expected, the total chlorophyll content in the treatments differed significantly during all of the phenological phases evaluated. It was found that height ($p < 0.001$), the cultivar ($p < 0.001$), and their interaction ($p < 0.05$) had a differential effect on the index (Table 3). It was identified that the highest chlorophyll content was present in the Blanca Real cultivar established in the temperate climate during the appearance of true leaves, Pasankalla in warm climate in the branching phase, and Pasankalla in cold climate during the last two phenological phases evaluated.



**Table 2.** Phenological performance of seven quinoa cultivars under three altitude gradients (days after planting).

| Altitudinal Gradient | Cultivar | First Pair of Leaves Visible (11) | One Side Shoot Visible (21) | Inflorescence Present (50) | Beginning of Anthesis (60) | Milky Grain (81) | Thick Grain (85) | Plant Dead and Dry (97) |
|---|---|---|---|---|---|---|---|---|
| Cold | Blanca Real | 13.66 ± 1.52 abcde | 21.66 ± 1.15 f | 50 ± 2 abcd | 87 ± 2 ab | 97.66 ± 2.51 abc | 111.33 ± 2.88 abc | 124.33 ± 4.04 bcd |
| | Pasankalla | 10.66 ± 0.57 bcdef | 32 ± 2 ab | 48.66 ± 1.15 abcd | 74.66 ± 1.52 fgh | 101.66 ± 3.21 ab | 110.66 ± 7.50 abc | 122.33 ± 2.08 bcde |
| | Nariño | 14.66 ± 1.15 abc | 23.66 ± 0.57 ef | 42.33 ± 2.51 fg | 75.33 ± 2.51 fgh | 86.33 ± 3.05 abc | 99 ± 7.81 bc | 111.33 ± 3.51 ef |
| | Soracá | 13.33 ± 1.52 abcde | 24.66 ± 2.51 cdef | 51.33 ± 1.52 abcd | 91.33 ± 1.52 a | 108.33 ± 5.50 a | 124.33 ± 12.89 a | 136.66 ± 1.52 a |
| | Titicaca | 14.33 ± 0.57 abcd | 33.66 ± 1.52 a | 51.66 ± 1.52 abc | 76 ± 1 efg | 87.33 ± 3.78 c | 100 ± 5 bc | 114 ± 5.29 def |
| | Nueva | 11 ± 1 bcdef | 23.33 ± 2.08 ef | 51 ± 1.15 abcd | 76.33 ± 1.52 efg | 86.66 ± 4.16 abc | 113.66 ± 5.13 ab | 111.33 ± 2.08 bcde |
| | Puno | 9.66 ± 1.52 cdef | 22.33 ± 1.52 ef | 47 ± 1.73 cdef | 77.66 ± 1.52 def | 89 ± 5.29 bc | 106 ± 5.19 bc | 116.33 ± 4.72 def |
| Temperate | Blanca real | 15.66 ± 2.08 ab | 24 ± 1 ef | 50 ± 2 abcd | 85 ± 1 bc | 96 ± 5.29 abc | 107.66 ± 2.51 abc | 118.33 ± 1.52 cdef |
| | Pasankalla | 16.33 ± 1.52 a | 31.33 ± 1.15 ab | 49.66 ± 1.52 abcd | 73 ± 1 fgh | 96.33 ± 5.50 abc | 109 ± 3.60 abc | 129.66 ± 1.53 abc |
| | Nariño | 13.33 ± 30.5 abcde | 24.33 ± 1.52 def | 43.33 ± 1.52 efg | 74 ± 1 fgh | 86.33 ± 5.50 c | 100.66 ± 6.65 bc | 113 ± 4.35 def |
| | Soracá | 14.33 ± 1.15 abcde | 28.66 ± 1.52 bcd | 51.33 ± 1.52 abcd | 83 ± 2.64 bc | 95 ± 5 abc | 112.66 ± 6.42 abc | 132 ± 2 ab |
| | Titicaca | 10.33 ± 1.52 cdef | 35.33 ± 1.52 a | 53 ± 1 ab | 75 ± 1 fgh | 87 ± 4.35 c | 101.33 ± 5.50 bc | 113.66 ± 5.05 def |
| | Nueva | 10 ± 1 cdef | 25 ± 1 cdef | 53.33 ± 1.52 a | 72.66 ± 1.52 fgh | 90 ± 2.64 bc | 104 ± 5.29 bc | 119.66 ± 1.52 cdef |
| | Puno | 9 ± 1 ef | 23.33 ± 1.15 ef | 48 ± 1 bcde | 76.33 ± 1.52 efg | 88 ± 2.64 bc | 102 ± 5.29 bc | 117.66 ± 2.51 def |
| Warm | Blanca real | 13.66 ± 2.08 abcde | 21.66 ± 1.15 f | 48.33 ± 1.52 abcde | 82.66 ± 1.52 bcd | 97 ± 3.60 abc | 107.33 ± 4.50 abc | 118.33 ± 1.52 cdef |
| | Pasankalla | 13.33 ± 1.52 abcde | 29 ± 1.73 bc | 50.33 ± 3.5 abcd | 70.66 ± 0.57 h | 93 ± 7 bc | 111 ± 3.60 abc | 118.66 ± 3.21 cdef |
| | Nariño | 11.66 ± 2.51 abcdef | 22.66 ± 1.15 ef | 41 ± 1 g | 71.66 ± 1.52 gh | 85 ± 4.35 c | 95.66 ± 4.93 c | 111.66 ± 7.63 ef |
| | Soracá | 12 ± 1 abcdef | 26.66 ± 1.52 cde | 49.33 ± 1.52 abcd | 81 ± 2.64 cde | 91 ± 4.58 bc | 109 ± 4.58 abc | 121 ± 3.60 bcdef |
| | Titicaca | 9.66 ± 2.08 cdef | 34 ± 1 a | 51.33 ± 1.15 abcd | 72.66 ± 1.52 fgh | 86.66 ± 4.16 c | 101 ± 2.64 bc | 111 ± 3.60 ef |
| | Nueva | 9.33 ± 1.52 def | 23 ± 1 ef | 51 ± 1 abcd | 71.33 ± 1.52 gh | 90.33 ± 4.35 bc | 102 ± 5.29 bc | 113 ± 4.35 def |
| | Puno | 8 ± 2 f | 21.33 ± 1.52 f | 46.33 ± 1.52 def | 73.33 ± 1.53 fgh | 85.33 ± 4.72 c | 96 ± 5.29 bc | 110 ± 4.58 f |
| Altitudinal | | ** | *** | * | *** | *** | ** | *** |
| Cultivar | | *** | *** | *** | *** | *** | *** | *** |
| A/C | | ** | NS | NS | * | NS | NS | * |

Results are expressed as the mean ± standard deviation ($n = 3$). Different letters indicate statistical differences, and asterisks indicate significant differences between treatments (* $p < 0.05$, ** $p < 0.01$, and *** $p < 0.001$) by Tukey's honestly significant difference (HSD) test.

**Table 3.** Total chlorophyll content (SPAD) in seven quinoa cultivars subjected to three altitude gradients.

| Altitudinal Gradient | Cultivar | First Pair of Leaves Visible (11) | First Pair of Leaves Visible (21) | First Pair of Leaves Visible (60) | Fruit Set (70) |
|---|---|---|---|---|---|
| Cold | Blanca Real | 28 ± 2.64 [bcd] | 40 ± 2 [bcde] | 46.66 ± 1.57 [cdef] | 40 ± 2 [ef] |
| | Pasankalla | 28.33 ± 3.78 [bcd] | 42 ± 1.73 [bcd] | 57.66 ± 2.51 [a] | 57.66 ± 2.51 [a] |
| | Nariño | 25 ± 3 [cde] | 37 ± 1.73 [de] | 50 ± 2 [abcde] | 32 ± 2 [g] |
| | Soracá | 20.66 ± 1.15 [e] | 34.66 ± 0.57 [e] | 56.66 ± 3.51 [ab] | 54.66 ± 5.03 [ab] |
| | Titicaca | 22.33 ± 2.08 [de] | 39.66 ± 1.52 [bcde] | 42.33 ± 2.51 [ef] | 31.66 ± 2.88 [g] |
| | Nueva | 22.66 ± 2.59 [de] | 34.66 ± 2.51 [e] | 48 ± 3 [cdef] | 40.33 ± 1.52 [ef] |
| | Puno | 21 ± 1 [e] | 40.33 ± 0.57 [bcde] | 48.33 ± 2.08 [cdef] | 36 ± 1.73 [fg] |
| Temperate | Blanca Real | 35 ± 1 [a] | 44.66 ± 1.52 [abc] | 45 ± 2 [ef] | 37.66 ± 2.51 [fg] |
| | Pasankalla | 25 ± 2.64 [cde] | 46 ± 5.29 [ab] | 54.33 ± 4.04 [abc] | 52.33 ± 2.51 [abc] |
| | Nariño | 26 ± 1.73 [bcde] | 39.66 ± 2.08 [bcde] | 47.66 ± 2.51 [cdef] | 31 ± 1 [g] |
| | Soracá | 26.66 ± 1.52 [bcde] | 39.33 ± 1.52 [bcde] | 53 ± 3.60 [abcd] | 50.33 ± 1.52 [bcd] |
| | Titicaca | 28 ± 2 [bcd] | 41.33 ± 3.51 [bcde] | 42.33 ± 2.51 [ef] | 31 ± 1 [g] |
| | Nueva | 26 ± 1.73 [bcde] | 39.66 ± 2.08 [bcde] | 46.33 ± 1.52 [def] | 37 ± 2 [fg] |
| | Puno | 30 ± 2 [abc] | 41.33 ± 1.52 [bcde] | 48.33 ± 2.51 [cdef] | 34 ± 2 [fg] |
| Warm | Blanca Real | 27 ± 2 [bcde] | 41.33 ± 1.57 [bcde] | 43.33 ± 2.08 [ef] | 32.66 ± 2.08 [g] |
| | Pasankalla | 32.33 ± 2.51 [ab] | 51 ± 1 [a] | 53 ± 2 [abcd] | 45 ± 3 [de] |
| | Nariño | 27.66 ± 2.08 [bcd] | 41 ± 1.73 [bcde] | 46.66 ± 1.52 [cdef] | 31 ± 1 [g] |
| | Soracá | 28 ± 1 [bcd] | 40 ± 2 [bcde] | 49.66 ± 2.51 [bcdef] | 46.66 ± 2.08 [cde] |
| | Titicaca | 26.66 ± 1.52 [bcde] | 38 ± 2 [cde] | 42 ± 2 [f] | 30.66 ± 1.15 [g] |
| | Nueva | 28.33 ± 1.15 [bcd] | 38.66 ± 2.51 [cde] | 44.66 ± 2.51 [ef] | 32.33 ± 2.51 [g] |
| | Puno | 28.66 ± 1.15 [abcd] | 41.33 ± 2.08 [bcde] | 48 ± 2 [cdef] | 33.66 ± 3.21 [fg] |
| Altitudinal | | *** | *** | *** | *** |
| Cultivar | | *** | *** | *** | *** |
| A/C | | *** | * | NS | ** |

Results are expressed as the mean ± standard deviation (*n* = 3). Different letters indicate statistical differences, and asterisks indicate significant differences between treatments (* $p < 0.05$, ** $p < 0.01$, and *** $p < 0.001$) by Tukey's honestly significant difference (HSD) test.

The total content of chlorophyll was higher in the 50% flowering phase for all treatments. However, a higher content was identified for the Soracá and Pasankalla cultivars, which reached values of up to 57 and 56 SPAD units, respectively, under cold weather conditions, levels similar to those reported by Garcia-Parra et al. [24], who found values of up to 66.4 SPAD units in the flowering phase in an altitudinal gradient above 2500 m (cold climate). In addition, a slight decrease in total chlorophyll content was observed in the flowering and grain-filling phases in all treatments. Such behavior was attributed to the increase in temperature due to the effect the altitude gradient, which was highly significant for this variable, since its incidence results in some notorious changes in photosynthetic activity and, therefore, in the formation and functionality of chlorophylls, giving rise to an accelerated rate of vegetative development as day and night temperatures increase [25].

It is worth noting that the cultivar factor is of high significance to the content of total chlorophyll, which is attributed to the influence of environmental temperature in each of the study areas, since it has been reported that this climatic element affects the photosynthetic apparatus of plants, mainly in the stability of the thylakoid membranes, given that their increase can cause denaturation of the proteins and phospholipids that make up the photosystems [26].

Another determining factor of the chlorophyll content in plants occurs at the beginning of the physiological maturity phase, since at this moment a marked accumulation of ethylene is generated, which favors the maturation of the tissues and, therefore, a significant increase in gerontoplasts that give slow step to expression of remaining carotenoids and anthocyanins in leaf and stem tissues [27].

### 3.3. Chlorophyll Fluorescence (Fv/Fm)

The maximum quantum efficiency of photosystem II is a variable that can be affected by different external factors, as manifested in this research. It was found that the altitude

gradient considerably influences the Fv/Fm ($p < 0.001$), while the cultivar factor and its interaction do not exert significant influence (Table 4). In addition, it was identified that cultivars with a productive tradition in the central region of Colombia (Soracá and Nariño) are more susceptible to changes in altitude gradient (temperate and warm climate), mainly in the phases of true leaves and grain filling.

**Table 4.** Effect of the altitudinal gradient and the cultivar on the Fv/Fm.

| Altitudinal Gradient | Cultivar | First Pair of Leaves Visible (11) | First Pair of Leaves Visible (21) | First Pair of Leaves Visible (60) | Fruit Set (70) |
|---|---|---|---|---|---|
| Cold | Blanca Real | 0.81 ±0.010 [abcd] | 0.77 ± 0.010 [abcd] | 0.76 ± 0.011 [ab] | 0.68 ± 0.015 [ab] |
| | Pasankalla | 0.82 ± 0.020 [abc] | 0.77 ± 0.020 [abc] | 0.76 ± 0.015 [ab] | 0.69 ± 0.010 [a] |
| | Nariño | 0.83 ± 0.010 [a] | 0.78 ± 0.015 [a] | 0.77 ± 0.010 [a] | 0.66 ± 0.011 [ab] |
| | Soracá | 0.82 ± 0.010 [abc] | 0.76 ± 0.010 [abcde] | 0.75 ± 0.020 [abcd] | 0.68 ± 0.015 [ab] |
| | Titicaca | 0.80 ± 0.005 [abcde] | 0.77 ± 0.010 [abcd] | 0.76 ± 0.010 [abc] | 0.67 ± 0.015 [ab] |
| | Nueva | 0.81 ± 0.010 [abcd] | 0.75 ± 0.005 [abcdef] | 0.78 ± 0.010 [a] | 0.67 ± 0.015 [ab] |
| | Puno | 0.82 ± 0.020 [ab] | 0.78 ± 0.005 [ab] | 0.77 ± 0.020 [a] | 0.69 ± 0.010 [a] |
| Temperate | Blanca Real | 0.79 ± 0.015 [abcdef] | 0.74 ± 0.011 [abcdef] | 0.72 ± 0.015 [bcde] | 0.65 ± 0.020 [ab] |
| | Pasankalla | 0.79 ± 0.010 [abcdef] | 0.72 ± 0.015 [def] | 0.71 ± 0.010 [e] | 0.64 ± 0.005 [b] |
| | Nariño | 0.79 ± 0.005 [abcdef] | 0.74 ± 0.017 [abcdef] | 0.72 ± 0.010 [cde] | 0.66 ± 0.015 [ab] |
| | Soracá | 0.76 ± 0.010 [f] | 0.74 ± 0.010 [abcdef] | 0.72 ± 0.015 [bcde] | 0.66 ± 0.005 [ab] |
| | Titicaca | 0.77 ± 0.015 [def] | 0.73 ± 0.025 [cdef] | 0.7 ± 0.015 [e] | 0.64 ± 0.005 [b] |
| | Nueva | 0.76 ± 0.015 [f] | 0.73 ± 0.025 [bcdef] | 0.71 ± 0.010 [e] | 0.65 ± 0.015 [ab] |
| | Puno | 0.78 ± 0.005 [bcdef] | 0.72 ± 0.017 [ef] | 0.7 ± 0.011 [e] | 0.66 ± 0.011 [ab] |
| Warm | Blanca Real | 0.76 ± 0.015 [ef] | 0.71 ± 0.015 [ef] | 0.7 ± 0.010 [e] | 0.65 ± 0.025 [ab] |
| | Pasankalla | 0.76 ± 0.011 [ef] | 0.71 ± 0.015 [ef] | 0.71 ± 0.020 [de] | 0.65 ± 0.020 [ab] |
| | Nariño | 0.78 ± 0.005 [bcdef] | 0.72 ± 0.010 [ef] | 0.71 ± 0.010 [e] | 0.64 ± 0.005 [b] |
| | Soracá | 0.78 ± 0.010 [cdef] | 0.73 ± 0.017 [cdef] | 0.71 ± 0.010 [bcde] | 0.64 ± 0.005 [b] |
| | Titicaca | 0.78 ± 0.020 [bcdef] | 0.71 ± 0.017 [f] | 0.71 ± 0.015 [e] | 0.65 ± 0.011 [ab] |
| | Nueva | 0.79 ± 0.005 [abcdef] | 0.71 ± 0.015 [ef] | 0.69 ± 0.011 [e] | 0.66 ± 0.020 [ab] |
| | Puno | 0.78 ± 0.015 [bcdef] | 0.71 ± 0.020 [ef] | 0.7 ± 0.010 [e] | 0.67 ± 0.005 [ab] |
| Altitudinal | | *** | *** | *** | *** |
| Cultivar | | NS | NS | NS | NS |
| A/C | | * | NS | NS | NS |

Results are expressed as the mean ± standard deviation ($n = 3$). Different letters indicate statistical differences, and asterisks indicate significant differences between treatments (* $p < 0.05$ and *** $p < 0.001$) by Tukey's honestly significant difference (HSD) test.

The activity of the fluorescence of chlorophyll was determined in the most important phases established by Bertero and Ruiz [28], which allows identifying the photochemical activity throughout the phenological cycle. During the study, it was observed that the environmental conditions have a marked influence on the maximum quantum efficiency of photosystem II, identified as a variable highly sensitive to genetic and environmental modifications. Bascuñán-Godoy et al. [29] reported variations in Fv/Fm due to the influence of the quinoa genotype used and the edaphic character. In addition, other studies have reported the effect of drought stress on Fv/Fm, causing a significant decrease [21].

However, it has been found that environmental temperature does not have a negative effect on the Fv/Fm indices, as has been reported in wheat and quinoa plants subjected to high temperatures (40–45 °C), since a slight decrease was observed in the index [30,31]. Therefore, it is noteworthy that the index values between 0.8 and 0.85 indicated that the plant was not under a state of stress [32,33], but that according to what was obtained in this research, it was possible to observe a slight reduction in Fv/Fm as the altitude gradient decreased.

In this way, it can be concluded that the general stability of Fv/Fm during the phases of true leaves, branching and flowering indicates that temperature is the main climatic variable of the altitudinal gradient, which does not affect the efficiency of electron transport in the photosynthetic apparatus and, therefore, photoinhibition is not generated, so the reduction in the values of this variable during the grain-filling phase is attributed to the

senescence of the foliage due to the end of the productive cycle, as demonstrated by Panda and Sarkar [34].

### 3.4. Stomatal Density

Another interesting observation in this study was the phenomenon generated by altitudinal gradient, cultivar and their interaction in the number of stomata per unit area (Figure 2). It was found that warm weather conditions influenced the number of leaf surface structures, while, in the case of the Blanca Real cultivar under warm weather conditions, the amount decreased by up to 25.6% compared to its production in temperate climate, which was similar for the Nariño, Nueva, Pasankalla and Titicaca cultivars, while in Puno and Soracá, the greatest number of stomata occurred in plants established in cold climate.

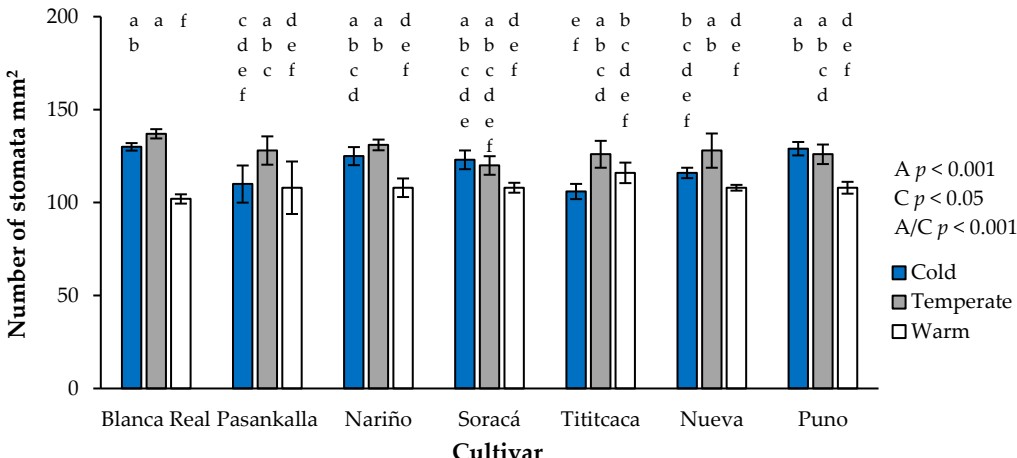

**Figure 2.** Stomatal density of different quinoa cultivars subjected to three altitude gradients. The bars indicate standard error. Different letters indicate significant differences. A: Altitudinal; C: Cultivar; A/C: Altitudinal/Cultivar.

Stomatal density varied significantly in the study as a response between cultivars, environments and their interaction. This phenomenon can be derived from factors such as the hydric status of the plant and the environmental temperature mainly, which may be the reflection of a marked reduction in precipitation during the months of July and August, which coincided with the phase in which the precipitation was measured, and agrees with the findings on stomatal density reported by Issa-Ali et al. [35], who reported constant variations in the number of stomata after inducing edaphic drought, which allows this variable to be recognized as a phenotypic plasticity response capable of determining transpiration in foliar tissues.

### 3.5. Specific Leaf Area

The specific leaf area presented high variation in relation to the cultivars, the altitude gradient and their combination (Figure 3). It was possible to identify a greater proportion of the specific foliar area in the cold region, with the exception of the cultivars Pasankalla, Blanca Real and Puno, and it was possible to determine a greater difference between the altitude gradients of cold and warm climate, mainly in the cultivars Blanca Real, Nueva, Titicaca and Soracá, which presented a reduction in this variable by 40.7%, 39.9%, 39% and 30.1%, respectively, as the altitude gradient was reduced.

Different researches have determined that the specific leaf area (SFA) can be understood as an indicator of photosynthetic activity and plant growth performance. The AFE showed significant changes in all cultivars as the altitudinal gradient was reduced, an aspect that has been observed in different grass species when the temperature of the production area increases [36]. In this same sense, it has been shown in rice plants (*Oryza sativa*) that the increase in daytime temperature reduces the AFE [37], most likely as a response to the loss of water in the tissue, which can happen more easily in areas with warmer conditions,

due to the insensitive meatiness of the leaf tissue that quinoa leaves have and the C3 metabolism that makes it difficult for the leaves to retain and fix $CO_2$ for gas exchange.

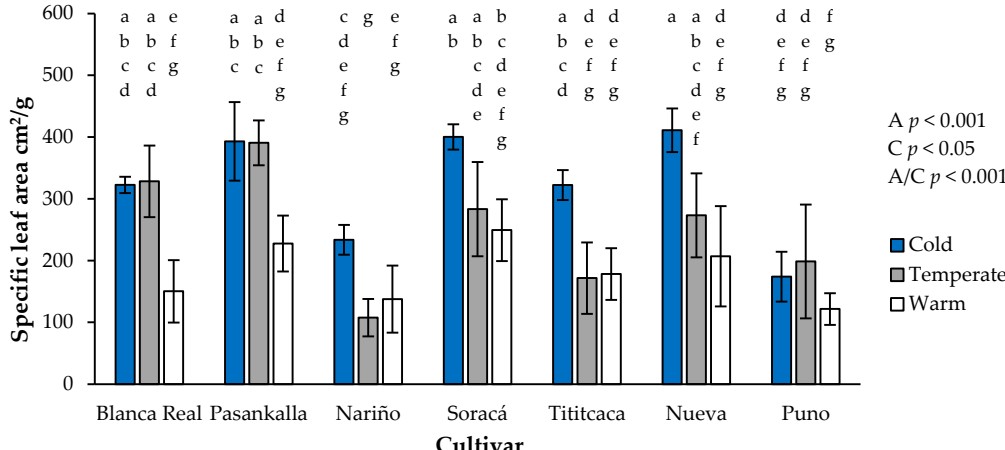

**Figure 3.** Specific leaf area performance of different quinoa cultivars subjected to three altitude gradients. The bars indicate standard error. Different letters indicate significant differences. A: Altitudinal; C: Cultivar; A/C: Altitudinal/Cultivar.

### 3.6. Grain Yield

Grain production per linear meter showed significant statistical differences (Figure 4), revealing a marked influence between the cultivars evaluated and the altitude gradient. The cultivar with the best productive performance was Soracá. However, its productivity was affected in lower altitude conditions, which reduced grain production by up to 60.1%, which was very similar to what happened with the Nariño and Nueva cultivars, where their production fell by 65% and 52.5%, respectively. Additionally, it was possible to identify a particular behavior for the Puno cultivar, which showed its best performance under altitude conditions with a temperate climate.

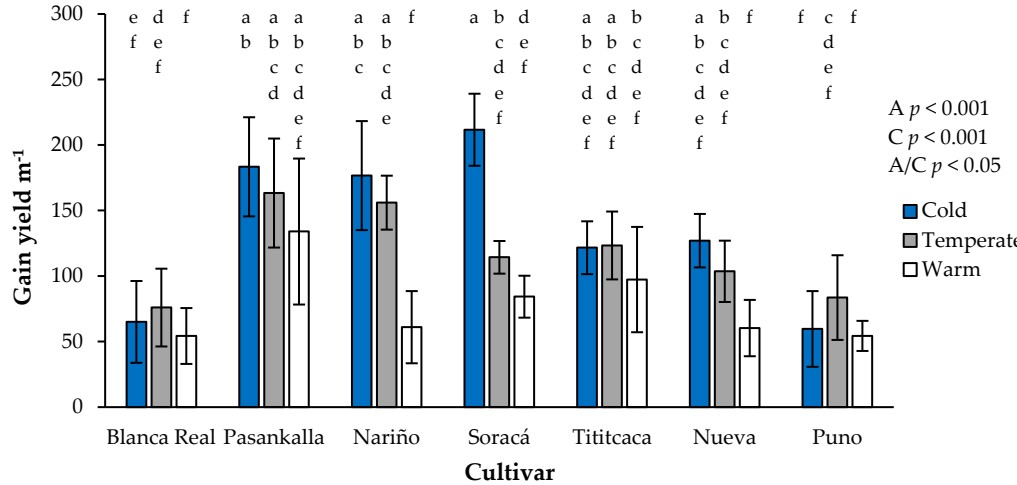

**Figure 4.** Grain yield of different quinoa cultivars subjected to three altitude gradients. The bars indicate standard error. Different letters indicate significant differences. A: Altitudinal; C: Cultivar; A/C: Altitudinal/Cultivar.

The grain yield varied significantly in relation to the treatment; in this sense, different studies have reported that productivity between quinoa cultivars is variable [38,39]. However, it has been found that there is no significant variation in grain production rates when quinoa plants are subjected to an increase in environmental temperature [40], but there

is an effect from the availability of water and nutrients in terms of the amount of grain harvested [41].

Some studies have reported a significant change in yield when the night temperature was increased in certain growth phases, causing a significant grain reduction between the cultivars evaluated and the phenological phase in which the heat treatment was carried out [25], an aspect that can be attributed to the accelerated use of the sugar reserve in organs that are considered landfills but do not benefit productivity.

### 3.7. Protein Content

The cultivars showed differences in grain protein concentration, highlighting the highest content in the Titicaca cultivar under cold and temperate climate conditions, while the lowest content was presented for the same cultivar under warm climate conditions (Figure 5). The reduction in the altitudinal gradient had a positive association with the protein content in the grain ($p < 0.001$). No association was found between the cultivars used and the altitudinal gradient ($p = 0.1$); in addition, it is important to note that the protein content showed highly significant differences between the cultivars evaluated.

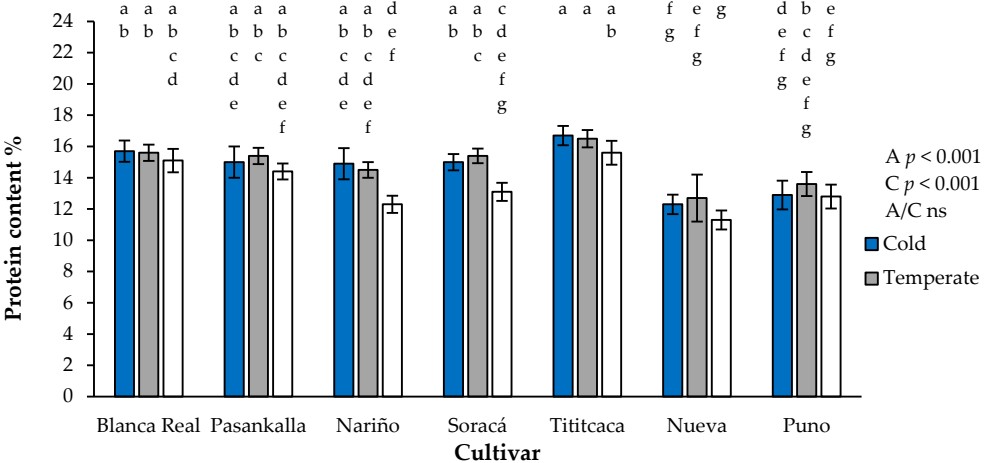

**Figure 5.** Protein content of different quinoa cultivars subjected to three altitudinal gradients. The bars indicate standard error. Different letters indicate significant differences. A: Altitudinal; C: Cultivar; A/C: Altitudinal/Cultivar.

The response of the compositional characteristics of the grains is the result of all of the physiological and biochemical performance of the plants. For this reason, research has been carried out in which the effect of the agroecological environment on the characteristics of quinoa grains was evaluated, as reported by Reguera et al. [2], who found significant statistical differences in the protein content of different quinoa cultivars planted under different agroclimatic conditions, in addition to identifying the highest percentage of protein in the Titicaca cultivar. In this sense, Gonzalez et al. [42], found significant differences between cultivars from different agroecological regions, which also varied in the profile of amino acids present in quinoa seeds.

This situation is a reflection of the influence of environmental conditions such as ambient temperature, hydric status and the physicochemical characteristics of the soil on the source-dump dynamics, while in many cases the induction of stress in the plants generates a benefit in the compositional characteristics of the seeds as a response to the accumulation of soluble sugars, amino acids and proteins [43].

### 3.8. Altitudinal Gradient Relationship with Seven Quinoa Cultivars

The treatments were grouped in the Bootstrap-based cluster analysis with respect to the physiological and compositional performance of the seeds (Figure 6A). Two macrogroups were evidenced. The first comprised the Soracá cultivar planted in the different climates,



Pasankalla and Blanca Real in cold and temperate, Nueva in cold and warm and Titicaca in cold, while the second group covered the other treatments. The Nariño cultivar under average climate conditions expressed an intermediate performance between the other treatments, which allowed them to be grouped according to the response that each of the cultivars evaluated under the three altitudinal gradients had, mainly due to physiological responses in the field. In addition, three groups of cultivars were identified that performed in a very similar way with regard to physiology, which could be corroborated with a starting value greater than 98%.

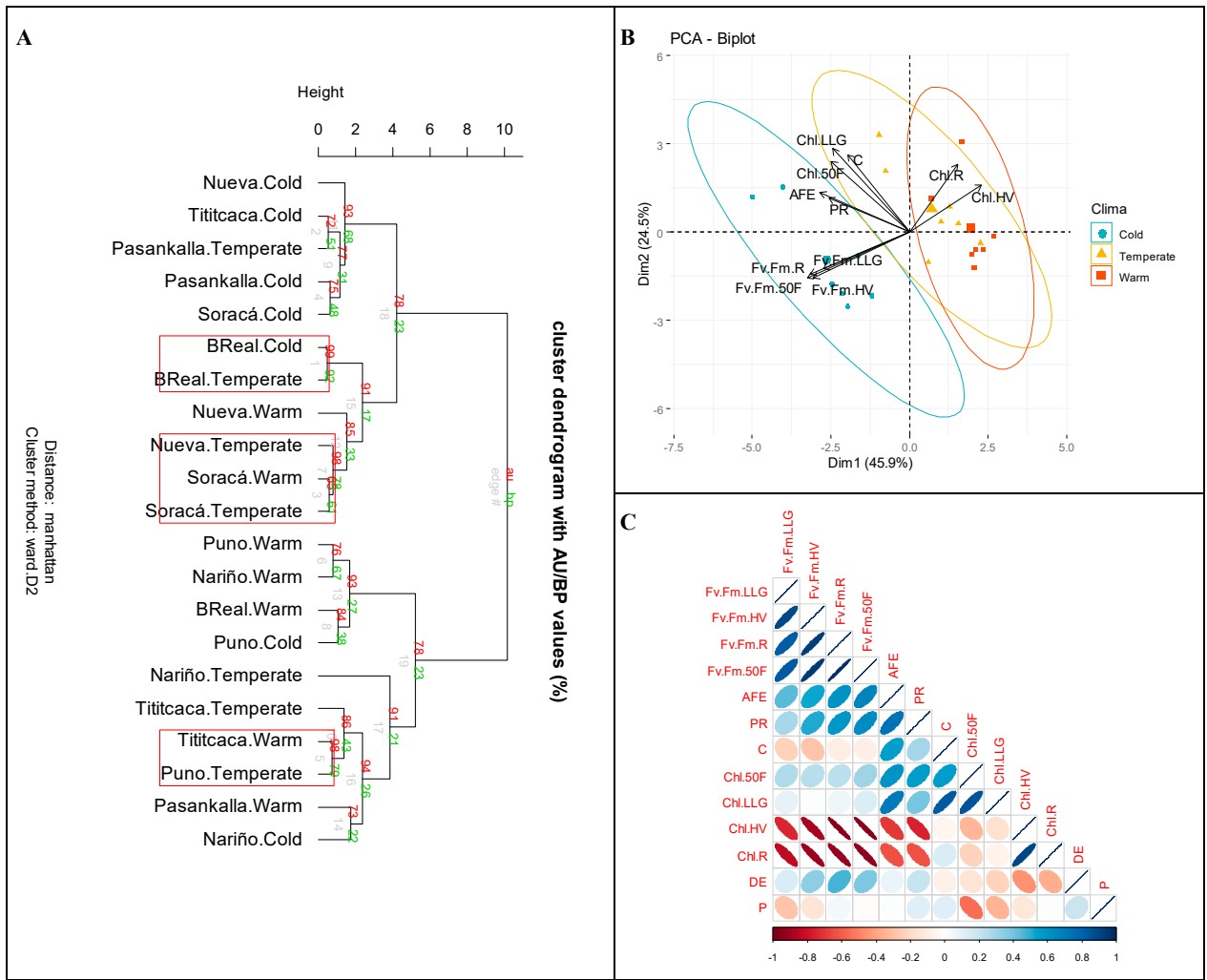

**Figure 6.** Grouping of treatments with respect to their physiological response. (**A**) Bootstrap-based cluster analysis. AU: Approximately Unbiased; BP: Bootstrap Probability (**B**) Principal component analysis of physiological variables with climate grouping. (**C**) Spearman correlation.

According to the principal component analysis PCA, it was found that the first component explained 45.9% of the variance and the second component, 24.5% (70.4% of the accumulated variance) (Figure 6B). In addition, it was identified that the warm climate treatments were similar to those of the temperate climate, while those of the cold climate had a greater association with the variables of the maximum quantum efficiency of the FSII. Time to harvest tended to increase in the temperate climate and decrease in the warm climate, while yield per plant and AFE were significantly higher in cool climate cultivars. It was possible to identify interaction between the measured variables using the Spearman correlation (Figure 6C), highlighting a high relationship between time to harvest, AFE and grain production, while stomatal density had a slight relationship with Fv/Fm variables.

However, this last variable presented a negative correlation with the chlorophyll content in the different phenological phases.

HV: true leaves, R: Branching, 50F: 50% flowering, LLG: grain filling, PR: grain production, AFE: specific leaf area, P: protein, Chl: chlorophyll content, Fv.Fm: maximum quantum efficiency of FSII, C: time to harvest, DE: stomatal density.

This panorama of similarities is the result of the genetic association that exists between quinoa cultivars and their attributes related to the different morphological characters of the leaves, stems and inflorescence, mainly [12,16]. Thus, the physiological response has been a set of species grouping characters, as has been shown in wheat (*Triticum aestivum* L.) [44], rice (*Oryza sativa* L.) [45] and maize (*Zea mays* L.) [46]. The Soracá and Pasankalla cultivars present high physiological variability, given that they have not been incorporated into breeding programs, which means that they present high variability in response to stress generated by temperature, radiation, salinity and water status [13], an aspect that allowed them to show homogeneity in the phenological variables evaluated.

In the case of the Blanca Real, Nariño, Titicaca and Nueva cultivars, there were relevant changes due to the effect of the genetic selection that has been carried out on these cultivars, so that a reduction in the genetic pool could reduce the adaptability strategies of this cultivar. species. In addition, slight changes were identified in the physiological responses related to the cultivars established in temperate and warm climates, which is attributed to the phenotypic plasticity that some quinoa cultivars can achieve more easily, mainly in aspects related to the increase in temperature [47].

In addition, the relationships that are generated between the variables are relevant. A very close relationship has been identified between the time to harvest and the chlorophyll content in the leaves, while as its concentration increases in the foliar tissues, the phenological period of the cultivar tends to be prolonged, as has been observed in the cultivar Soracá [15]. In addition, the low variability in the maximum quantum efficiency of the FSII has been common in the vast majority of quinoa cultivars [31], so this variable must be combined with more response variables in order to establish more accurate results for physiological stress.

## 4. Conclusions

The edaphoclimatic conditions of the three altitude gradients had a differential effect on the phenological and physiological performance of the different quinoa cultivars evaluated, which allows us to highlight that the edaphoclimatic conditions of the production area could significantly alter the production parameters of quinoa, also modifying the compositional characteristics of its seeds. Although not all of the parameters evaluated varied to the same extent, this research corroborates that both the cultivar and the altitudinal gradient determined the productivity of the cultivar, but that the gradient and the cultivar, as independent factors, affected the protein content in the seeds. In addition, it is highlighted that the maximum quantum efficiency of photosystem II was the variable that varied the least, since its index did not change among the quinoa cultivars evaluated. However, the altitude gradient associated with changes in environmental temperature influenced the fluorescence of chlorophyll dynamics, manifesting a higher level of physiological stress in quinoa plants grown in temperate and warm climates.

**Author Contributions:** Conceptualization: M.G.-P. and D.R.-A.; methodology: M.G.-P.; software: M.G.-P. and D.R.-A.; validation: J.E.B.-G. and D.R.-A.; formal analysis: M.G.-P. and J.E.B.-G.; investigation: M.G.-P. and D.R.-A.; resources: M.G.-P., D.R.-A and J.E.B.-G. All authors have read and agreed to the published version of the manuscript.

**Funding:** The authors express gratitude to the Minciencias (Ministerio de Ciencia and Tecnología e Innovación) No. 779/2017. We are also grateful to the Boyacá Department of the government and the Universidad del Cauca, ID 5637 and ID 4854 project.

**Institutional Review Board Statement:** Not applicable.

**Informed Consent Statement:** Not applicable.

**Data Availability Statement:** The data generated within this work are open access and available to be shared with interested persons.

**Acknowledgments:** The authors thank Roman Stechauner-Rhoringer.

**Conflicts of Interest:** The authors declare no conflict of interest.

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
