# Peer review of "Effect of the Altitude Gradient on the Physiological Performance of Quinoa in the Central Region of Colombia"

_agronomy, doi:10.3390/agronomy12092112_

Round 1

Reviewer 1 Report

The submitted manuscript deals with the issue of the relationship between yield and its quality within growing areas. Quinoa is one of the many traditional Andean crops that has worldwide application. The results of the mentioned research can also be used in practical applications. The manuscript is written relatively carefully, even so I have a few notes and comments about it. Considering the practical impact, I would supplement the abstract with the division of the monitored cultivars into groups, as stated in the results. I would add the summary in the same way. I would add the course of the weather in the monitored areas to the methodology. The authors studied not only the influence of genotype, developmental stage and altitude on physiological and yield parameters, but unfortunately they only compared individual results separately, not in correlation, which is somewhat detrimental to the article. First of all, it concerns the number of stomata, where there is no evaluation in relation to photosynthesis, water regime, etc. There is no direct relationship between stomata and the content of pigments or their fluorescence. It is a pity that the SPAD method was used, where it is not possible to separate chlorophylls and carotenoids. Their ratio and quantity would certainly be interesting and it would also detect the transition to the senescence phase. I would appreciate the addition of photos or micrographs of the stomata. For graph 3, it is necessary to fix the legend of the y-axis, and graph No. 5 is more difficult to read. The discussion is sometimes rather descriptive. It is necessary to correct the citation in the list of used literature, as it is not uniform.

Author Response

We appreciate the corrections proposed by the respected reviewer.

  1. The correction suggested in the abstract of the paper was made.
  2. We built the diagram of the climatic conditions of the study areas.
  3. We add in the correlation graph the name of the stomatal density.
  4. Unfortunately, we could not carry out an evaluation of the photosynthetic activity since we do not have the equipment to do so.
  5. We adjust the names of the graphs
  6. Correction of the bibliography was made
  7. We corrected the words that were misspelled
  8.  
  9.  

Reviewer 2 Report

The paper is interesting and performs an in-depth study of a crop which is gaining interest in the western world such as quinoa (it was widely known in the Andean region since ancient times).

The study merits publication in Agronomy,  I have found only some minor points.

line 37: edaphoclimatic

Line 42: delete "." after industry.

Line 110: greenness

figure 5: align the red boxes.

Author Response

We appreciate the corrections proposed by the respected reviewer.

  1. The correction suggested in the abstract of the paper was made.
  2. We built the diagram of the climatic conditions of the study areas.
  3. We add in the correlation graph the name of the stomatal density.
  4. Unfortunately, we could not carry out an evaluation of the photosynthetic activity since we do not have the equipment to do so.
  5. We adjust the names of the graphs.
  6. Correction of the bibliography was made
  7. We corrected the words that were misspelled
